# An Angiogenic Gene Signature for Prediction of the Prognosis and Therapeutic Responses of Hepatocellular Carcinoma

**DOI:** 10.3390/ijms24043324

**Published:** 2023-02-07

**Authors:** Hongfei Ci, Xufeng Wang, Keyu Shen, Wei Du, Jiaming Zhou, Yan Fu, Qiongzhu Dong, Huliang Jia

**Affiliations:** 1Department of General Surgery, Huashan Hospital, Fudan University, Shanghai 200040, China; 2Key Laboratory of Whole-Period Monitoring and Precise Intervention of Digestive Cancer, Shanghai Municipal Health Commission (SMHC), Institute of Fudan-Minhang Academic Health System, Minhang Hospital, Fudan University, Shanghai 200437, China

**Keywords:** hepatocellular carcinoma, bioinformatics, prognostic prediction model, angiogenesis effect

## Abstract

Among cancer-related deaths worldwide, hepatocellular carcinoma (HCC) ranks second. The hypervascular feature of most HCC underlines the importance of angiogenesis in therapy. This study aimed to identify the key genes which could characterize the angiogenic molecular features of HCC and further explore therapeutic targets to improve patients’ prognosis. Public RNAseq and clinical data are from TCGA, ICGC, and GEO. Angiogenesis-associated genes were downloaded from the GeneCards database. Then, we used multi-regression analysis to generate a risk score model. This model was trained on the TCGA cohort (n = 343) and validated on the GEO cohort (n = 242). The predicting therapy in the model was further evaluated by the DEPMAP database. We developed a fourteen-angiogenesis-related gene signature that was distinctly associated with overall survival (OS). Through the nomograms, our signature was proven to possess a better predictive role in HCC prognosis. The patients in higher-risk groups displayed a higher tumor mutation burden (TMB). Interestingly, our model could group subsets of patients with different sensitivities to immune checkpoint inhibitors (ICIs) and Sorafenib. We also predicted that Crizotinib, an anti-angiogenic drug, might be more sensitive to these patients with high-risk scores by the DEPMAP. The inhibitory effect of Crizotinib in human vascular cells was obvious in vitro and in vivo. This work established a novel HCC classification based on the gene expression values of angiogenesis genes. Moreover, we predicted that Crizotinib might be more effective in the high-risk patients in our model.

## 1. Introduction

HCC, the most familiar primary liver cancer, is the main common cancer worldwide [1]. The incidence of HCC had an increase by 75% from 1990 to 2015, as a result of the combination of aging, population growth, and metabolic syndrome. Asia has the highest burden of liver cancer incidence, death, and years of life lost [2]. It is reported that 72% of HCC cases occur in Asia (more than half in China) [3]. In the past decades, great efforts have been made in deciphering the epidemiology, risk factors, and molecular and genetic profiles of HCC‚ preventing, monitoring, early diagnosing, and treating HCC [4]. Vascular abnormality is a general hallmark of solid tumors. Therapeutic approaches to normalize blood vessels have been approved for first-line therapy, but limited efficacy and resistance pose hard problems. Sorafenib is the first FDA-approved angiogenesis inhibitor for the first-line treatment of advanced HCC [5]. Subsequently, several other angiogenesis inhibitors (Tyrosine kinase inhibitors, TKI) were developed to overcome the drug resistance and improve the therapeutic efficacy [6]. The treatment response may vary from person to person, so identifying the biomarkers of treatment response will facilitate the development of personalized treatment strategies and improve the outcome for patients.

RNA sequencing (RNAseq) is an omnipresent tool in deciphering the molecular mechanism of cancer progression [7]. RNA-seq-generated gene expression data and clinical information of different tumors have been stored in some public databases, such as The Cancer Genome Atlas Program (TCGA), the Gene Expression Omnibus (GEO) and the International Cancer Genome Consortium (ICGC) [8]. Systematic analysis of RNA-seq and clinical materials of cancer patients using this database will help identify gene expression signatures suitable for the estimation of prognosis and the indication of individualized therapy in HCC.

In this work, our aim is to identify the potential value of angiogenesis-related prognostic genes (ARPGs) in predicting the prognosis and the outcome of HCC patients.

## 2. Results

### 2.1. Construction of the Prognostic Model with the Angiogenesis-Related Gene in HCC

The first step in our analysis was to perform univariate Cox regressions on the TCGA and ICGC cohorts. A total of 3101 and 917 genes were identified as HCC survival-related genes in TCGA and ICGC cohorts, respectively. After intersecting with the angiogenesis-related gene set, 32 angiogenesis-related genes were identified to be associated with the prognosis of HCC (Figure 1A). We then used LASSO regression analyses to screen the 29 feature genes (Figure 1B,C). After the multivariate Cox regression analysis using these 29 genes in the TCGA cohort, we identified 14 ARPGs as the best characteristic gene set and showed the hazard ratio values (Figure 1D). Through the HPA database, the levels of *TPX2*, *KIF4A*, *BCAR3*, and *YWHAG* are higher in HCC tissues, whereas the levels of *RAB11A*, *GOT2*, *KDR*, *ATF3*, and *SLC22A1* are higher in normal liver tissues (Figure 1E). The expression levels of 14 ARPGs in 22 HCC cell lines were visualized by a heatmap (Appendix A). We also tested the mRNA levels of two HCC cell lines by qRT-PCR (Appendix A). From these two figures, it can be seen that the expression levels of these genes in HCC cells are in good agreement. A heatmap of the TCGA-LIHC cohort was drawn to test the expression levels of 14 ARPGs in tumor tissue (Appendix A). 

### 2.2. Validation of the Prognostic Signature

For results confirmation, the model was further validated using a cohort of independent individuals. A Kaplan-Meier survival analysis revealed that samples with low-risk scores had longer OS than those with high-risk scores in both the TCGA training (Figure 2A) and GSE14520 validation samples (Figure 2B). The ARPGs risk score model was tested using time-dependent ROC curves and Area Under Curves (AUCs) during training and validation (Figure 2C,D). The AUCs curve exceeded 0.76 in the TCGA dataset and 0.63 in GSE14520 set at 1, 3, and 5 years. It was suggesting a favorable and stable predictive value of the model in the whole follow-up of patients. Following that, we ranked both the training and validation sets based on their risk scores (Figure 2E,F). The results suggested that HCC patients owning higher scores would get a worse OS.

### 2.3. Prognostic Nomogram of Risk Model by This Gene Set

We constructed a nomogram to conclude the fourteen markers for calculating the ARPG risk score easily and predict the 1-, 3-, and 5-year OS of HCC roughly (Figure 3A). The results exhibited that the nomogram-predicted 1-, 3-, and 5-year OS closely matched the best prediction performance (Figure 3B–D). We compared the expression levels of this 14-ARPG between normal and HCC tissues. The boxplot (Figure 3E) hinted that they might also serve as a diagnostic biomarker for HCC.

### 2.4. Comparison of High- and Low-Risk Groups for Somatic Variants

There are no identified biomarkers that can be used as predictive molecules for sensitivity to immunotherapy. TMB has the potential to respond well to immunotherapy. There is some research showing that first-line nivolumab combination with ipilimumab prolonged progression-free survival (PFS) compared to conventional chemotherapy for non-small cell lung carcinoma (NSCLC) patients with a high mutational burden, regardless of PD-L1 expression level [9]. Consequently, the more mutations there are, the higher the chance that some of the neo-antigens will be immunogenic, which is why T cells clear them [10]. We first studied the differences between the two groups in the top eight genes with the most frequent mutation rates in liver cancer (Figure 4A). The variant classification was also summarized with stacked bar plots and box plots representing variant types and variant numbers. (Figure 4B). The missense mutation ranked first in all variant classifications and the incidence of single nucleotide polymorphism (SNP) is much higher than that of other types of mutations in HCC. By a violin plot we could find that the number of TMB in the high risk-group is higher than that in the low-risk group (Figure 4C). The top six differentially mutated genes between the two groups were *CTNNB1*, *P53*, *TTN*, *ALB*, *MUC16*, and *PCLO* (Figure 4D). In the low-risk group with a good outcome, *CTNNB1* mutations were more prevalent, and several studies have shown that *CTNNB1*-mutated HCCs were well differentiated [11]. *TP53* has been identified as a biomarker of certain molecular characteristics and a prognostic factor for unfavorable survival in HCC by quite a few studies. *TP53* mutations were more frequent in individuals at high risk, based on our results (Figure 4D). Data from the mutational analysis suggested that high-risk patients might be better candidates for immunotherapy. 

### 2.5. Analyzing the Model’s Functional Enrichment

By dividing the 343 TCGA samples according to their median risk scores, two groups were formed. A volcano plot was made to show all differentially expressed genes (DEGs) between these two groups. As a result, we found that 1077 genes were upregulated in the high-risk group, whereas 275 genes were significantly downregulated (Figure 5A). To better understand the regulating effects of the 14-ARPG, we obtained a protein-protein interaction (PPI) network in Metascape (https://metascape.org/, accessed on 1 January 2022) by inputting the top 500 genes of upregulated genes, which did not include the 14 ARPGs. Furthermore, we extracted the five most meaningful MCODE components from the PPI network (Figure 5B), GPCR ligand binding, snRNA processing, and voltage-gated Potassium channels that were related to biological function. To explore the potential biological processes associated with the 14-ARPG, GO analyses were conducted using these top 500 DEGs (Padj ≤ 0.05) (Figure 5C). Considering the 14-gene signature in conjunction with some important signaling pathways, GSEA analysis was applied to find the related pathways. The Reactome pathway database enrichment results demonstrated that eukaryotic translation elongation, resolution of sister chromatid cohesion, kinesins, eukaryotic translation initiation, activation of the pre-replicative complex, and unwinding of DNA were significantly enriched (Figure 5D). In summary, we detected biological differences mainly associated with cell cycle-related pathways between high- and low-risk groups.

### 2.6. The Gene Signature for Nomogram Construction 

According to the above study, the risk score of our model and HCC patient prognosis are significantly correlated. In HCC patients, clinical factors, such as AJCC stage and age, are still important predictors of OS. So, we made a nomogram by integrating these classical risk factors with our 14-ARPG to develop an efficient method to predict the OS of HCC (Figure 6A). A patient’s overall score could be calculated by summing up all points of these variables, based on which the 1-, 3-, and 5-year subsistence rate would be estimated. We also constructed a ROC curve analysis for one year to compare our model with other traditional clinical data (Figure 6B). Our model has an AUC of 0.770, which is significantly higher than the others. We validated the nomogram by three calibration curves, where the 45° line represented the best prediction. Calibration curves indicated that our nomogram predicted well in 1-, 3-, and 5-years OS of HCC (Figure 6C–E). Exploring the association between the risk score and the clinical data, we found that the risk score varied significantly among HCC patients with different tumor stages (Figure 6F,G). Patients with HCC might be able to predict their tumor stage using the ARPGs’ risk score.

### 2.7. Distinct Sensitivity to Targeted Therapies and Immunotherapy for Different Groups

After validating the performance of the model in predicting diagnosis and prognosis, we attempted to predict treatment options for high-risk patients. GSE109211 contains 21 Sorafenib treatment responders and 46 non-responders who received Sorafenib, and their risk scores were calculated to examine whether the Sorafenib response was related to their risk signatures. There are 8 genes in our model differently expressed between the two groups (Figure 7A). Additionally, in responders to Sorafenib treatment, risk scores were significantly higher than in non-responders (*p* ≤ 0.001). This suggested that HCC patients with high-risk scores might be more sensitive to Sorafenib (Figure 7B). The effect of ICIs is tightly restricted by surface expression level, so we analyzed the association between the risk score and the treatment efficacy of immune checkpoint inhibitors using the TCIA database (https://tcia.at/home, accessed on 1 January 2022) [12,13]. The IPS developed from a four kinds immune cells-based panel of immune-related genes, has predictive value in patients treated with the CTLA-4 and PD-1 inhibitors. The low-risk group had significantly higher IPS and CTLA4 scores (*p* ≤ 0.05), providing evidence that LIHC samples with a low-risk score may be suitable for immunotherapy (Figure 7C,D).

### 2.8. Investigation of Potential Drugs Based on the Model

In the next step, we looked for potential drugs that might be suitable for high-risk patients. After estimating the AUC of 1448 compounds in the TCGA cohort by the DEPMAP database, we did a correlation analysis between the risk score and AUC. Then we summarized the top ten compounds exhibiting the strongest negative correlation with the risk score (Figure 8A). Crizotinib, an anti-cancer medication through inhibiting anaplastic lymphoma kinase (ALK) and c-ros oncogene 1 (ROS1), has already been approved for the treatment of NSCLC. We also compared the distribution of Crizotinib predicting AUC in different groups, and found that the overall sensitivity of the high-risk group to Crizotinib was higher (Figure 8B). We next confirmed the anti-migratory ability of Crizotinib in HUVECs by transwell assay (Figure 8C). We observed that Crizotinib decreased the tube formation capacity of HUVECs (Figure 8D). In addition, Crizotinib could increase the level of apoptosis in PLC/PRF/5 and MHCC97H cell lines (Figure 8E). Then the subcutaneous tumor models were established and treated with Crizotinib in C57 mice. Crizotinib could significantly inhibit tumor growth rate and weight (Figure 9A,B). IHC staining of the subcutaneous tumor showed that the proliferation marker KI67 and the vascular marker CD31 were significantly inhibited by Crizotinib (Figure 9C).

## 3. Discussion

Tumor angiogenesis is one of the special features of HCC progression, and many treatments for HCC revolve around the inhibition of angiogenesis and vascular normalization [14]. Aberrant microvasculature generally may seem arterialized, capillarized, and less dense than normal liver tissue [15]. Hypoxia is a characteristic of highly angiogenic cancer, such as HCC [16]. Hypoxia may promote HCC by reinforcing the abilities of invasion, progress, and resistance to therapies [17]. On the contrary, inducing vessel normalization and relieving hypoxia would delay the growth of cancer [18]. Because vascular abnormality is a hallmark of most solid cancers and facilitates immune evasion, combining ICIs and antiangiogenic therapies for HCC might improve the effectiveness of treatment and diminish the risk of drug adverse effects [19]. Lenvatinib is a novel TKI, with more potent activities against VEGF receptors and the FGF receptors’ family. Lenvatinib is an emerging antiangiogenic agent for first-line therapy in HCC [20]. Recently, some clinical trials and lab research reported the astonishing outcomes of combining anti-PD1 and Lenvatinib in HCC [21,22,23,24]. So, antiangiogenic therapy combined with ICIs may potentially improve HCC patient outcomes. 

Our study tried to find out a new angiogenic gene signature to filter high-risk HCC patients. There are 32 genes characterized as ARPGs in HCC. After building some regression models among them, we further identified that six genes (*TPX2*, *KIF4A*, *UBE2S*, *RAN*, *BCAR3*, *YWHAG*) are associated with significantly shorter OS whereas eight genes (*RAB11A*, *GPLD1*, *EPAS1*, *GOT2*, *KDR*, *ATF3*, *KLKB1*, *SLC22A1*) correlated with longer OS in HCC patients. As we know, gene expression levels and protein levels may not always correspond to each other, we identified ARPGs at a transcriptional level, providing clues for proteomics, metabolomics, and functional analysis in the future.

It has been reported that these genes are involved in the development of HCC in our model through biology experiments. Focusing on the six genes that strengthened the malignant phenotype in HCC (*TPX2*, *KIF4A*, *UBE2S*, *RAN*, *BCAR3*, *YWHAG*), we found that they are all associated with the PI3K/AKT pathway. Angiogenesis is a cooperative process between vascular endothelial cells and other cells, including the solid tumor cells, platelets, and immune cells, which would generate angiogenic factors and contribute to vascular tube formation [25]. The effects of antiangiogenic inhibitors in the PI3K pathway have been certified in many preclinical cancer models [26,27]. TPX2 plays an important role in spindle orientation and is frequently upregulated in cancer [28]. Silencing TPX2, the expression of PI3K, phospho-AKT, Bcl-2, c-Myc, and Cyclin D1 was decreased in HCC, obviously [29]. KIF4A is associated with chromosomes at the whole mitosis, and KIF4A downregulation is related to the inhibition of Akt kinase activity and intrinsic apoptosis signaling pathway activation [30,31]. RAN is a protein regulating the rate of nucleocytoplasmic shuttling, and cancer possessing KRAS activating mutation, c-Met enhancement, and PTEN-deletion are more sensitive to apoptosis induced by silencing RAN [32,33]. PI3K activation could promote BCAR3-mediated Rac activation due to PIP3 production by PI3K activating Rac/Cdc42 GEFs with pleckstrin homology domains [34]. YWHAG encodes a protein as a scaffold in multiprotein complexes in cancer cellular processes. It might promote the proliferation, migration, and invasion of gastric carcinoma by activating the RAS pathway [35]. With the approval of PI3K inhibitors for haematological malignancies, the approval for breast cancer has heralded a new era in the development of the PI3K drug [36]. But the effect of PI3K inhibitors in HCC remains largely unknown. As for the genes associated with better prognosis, some of them were researched before. It was found that EPAS1 could induce HCC apoptosis by regulating the TFDP3/E2F1 pathway [37]. As a membrane ion transporter, the knockdown of SLC22A1 has been found to trigger Sorafenib resistance in HCC [38]. It may be used as an important marker to predict the prognosis and therapeutic effect of liver cancer. The transcription factor ATF3 plays a key role in HCC apoptosis induced by niclosamide [39]. Inhibition of GOT2 can activate the PI3K/AKT/mTOR pathway to participate in glutamine metabolism and promote the progression of HCC [40].

Although some studies have built the prognosis model of HCC, our study tried to build a novel prognostic model in HCC by integrating fourteen angiogenesis-related genes. By bioinformatics analysis, our model showed some clinical values. By our model, patients in the high-risk group differed from those in the low-risk group in terms of mutation profile and sensitivity to ICIs. Further, we examined the relationship between gene signatures and prognosis in HCC patients, as well as predicted responses to various compounds, which might treat these patients with high risk. However, we believed that more research should be emphasized in subsequent studies to validate our model.

## 4. Materials and Methods

### 4.1. Sample and Data Collection

For the 373 liver hepatocellular carcinoma (LIHC) samples, corresponding RNA expression profiles were acquired from TCGA (https://cancergenome.nih.gov/, accessed on 1 January 2022). At the same time, their clinical data were collected. Our study only enrolled samples with a follow-up of more than 30 days. At the same time, the ICGC database (https://icgc.org/, accessed on 1 January 2022) was used to obtain clinical information and RNAseq of validated cases. The GSE14520 cohort (https://www.ncbi.nlm.nih.gov/geo/, accessed on 1 January 2022), including 242 HCC samples, was downloaded as the independent verification set. A gene set containing 4942 angiogenesis-related genes was downloaded from the GeneCards (https://www.genecards.org/, accessed on 1 January 2022) [41]. The expression profiling of 67 Sorafenib-treated HCC patients of GSE109211 was obtained from the Gene Expression Omnibus database (GEO), and there were 46 Sorafenib treatment nonresponders and 21 responders in these data. The Human Protein Atlas (HPA, https://www.proteinatlas.org/, accessed on 1 January 2022) is a database including immunohistochemistry data. We used the HPA to test the protein levels in HCC tissues and normal tissues. 

### 4.2. Establishment and Validation of a Prognostic Model

For confirming the underlying prognosis-related genes, we performed an analysis called univariate Cox regression for all genes in TCGA and ICGC cohorts. In addition, genes that were significantly associated with prognosis (*p* ≤ 0.05) were marked as prognosis-related genes in HCC. The genes overlapped in prognosis-related genes and angiogenesis-related genes were identified as potential ARPGs. Then, the LASSO regression method was performed to obtain the best ARPGs. Finally, we applied multivariate Cox regression analysis, identified 14 ARPGs, and constructed 14 gene signatures. In the next analysis, we added the risk score for everyone using the regression coefficients of ARPGs in the signatures and the corresponding expression values of ARPGs in the training and test sets. The sample risk score was counted using the below formula:*Riskscore* = ∑*iCoefficient* (*i*)∗*Expression of gene*(*i*)(1)

HCC patients in the TCGA database and GSE14520 cohort were grouped into the high-risk and low-risk groups on the strength of the median score. High-risk and low-risk groups were compared using the survival analysis. Additionally, the receiver operating characteristic (ROC) curve was utilized to test the precision of the prediction model.

### 4.3. Construction and Evaluation of a Predictive Nomogram

By integrating multiple risk elements, the nomogram is a great tool for quantifying the risk on individuals in a clinical setting. By the R package ‘rms’, we got a predictive nomogram with corresponding calibration curves from the independent predictive factors’ ARPGs. When the calibration curve comes near the 45-degree line, which is on behalf of the best prediction, the nomogram is better at prognosticating.

### 4.4. TMB Calculation

In the analysis of the genome, TMB was on behalf of the number of coding, somatic, indel mutations, and base substitutions for each megabase. In our study, we calculated TMB by R package ‘maftools’ to detect the mutation of the model [42].

### 4.5. Protein-Protein Interaction and Function Enrichment Analysis

The Metascape web tool facilitates multi-platform OMICs data analysis and accounts for experimental researchers through an integrated and user-friendly web interface [43]. The R package ‘clusterProfiler’ was utilized for Gene Ontology (GO) and Gene Set Enrichment Analysis (GSEA) pathway enrichment analysis to reveal their potential functions by comparing high-risk and low-risk groups [44].

### 4.6. Immune Status Analysis and Treatment Prediction 

We searched the keyword of ‘immune checkpoint gene’ in the PUBMED database (https://pubmed.ncbi.nlm.nih.gov/, accessed on 1 January 2022), and integrated the outcome into an immune checkpoint gene set. Immunophenoscores (IPSs), (containing IPS, IPS-CTLA4, IPS-PD1-PD-L1-PD-L2, IPS-PD1-PD-L1-PD-L2-CTLA4 scores) are downloaded in The Cancer Immunome Database (TCIA, https://tcia.at/home, accessed on 1 January 2022). Based on cell-type gene expression z-scores, IPSs assess the effectiveness of immune checkpoint inhibitors in different risk groups.

### 4.7. Prediction of Potential Therapeutic Agents in Patients with HCC

By the ‘oncoPredict’ R package and the sensitivity data of DEPMAP (https://depmap.org/portal/, accessed on 1 January 2022), we predicted some potential therapeutic agents for ARPGs’ high-group patients, and the smaller predicted AUC of the compound represented the more sensitive to the sample. First, high-risk and low-risk score TCGA samples were selected for differential groups for analysis, and we used the log2foldchange to identify differences.

### 4.8. Cells and Animals

Human liver cancer cell lines (MHCC97H and PLC/PRF/5), mouse HCC cell line (Hepa1-6), and human umbilical vein endothelial cells (HUVECs) were purchased at the Institute of Biochemistry and Cell Biology, Chinese Academy of Science (Shanghai, China). All cell lines were cultured at 37.0 °C and raised in a cell incubator with 5% CO_2_.

The male mice (C57BL/6, 6–8 weeks) were obtained from the Shanghai Slac Laboratory Animal Co. and fed with standard rodent chow and water. Before every surgery, all mice were given an injection of Pentobarbital Sodium (40 mg/kg) and conducted into the SPF lab. The Animal Care and Use Committee of Fudan University (Shanghai, China) approved all research involving animals in this work.

### 4.9. Migration Assay

The transwell assay involves seeding 2 × 10^5^ cells/well (three replications every group) in no-serum medium (Grand Island Biological Company, Grand Island, NE, USA) into the top chamber of transwell 24-well plates (Corning) with 8 μm pore filters. After that, the lower chamber was filled with the complete medium (including 20% FBS). After 72 h, washing the above surface of the filter membranes, crystal violet (0.5%) was used to stain the migrated cells on the below surface. Optical microscopy(40×) was used to assess migration levels.

### 4.10. Tube Formation Assays

A 96-well plate was filled with 50 μL of growth factor-reduced Matrigel basement membrane matrix (Corning Corporation). Next, we counted 3 × 10^4^ HUVECs and added them on top of the Matrigel matrix with DMSO or Crizotinib (50 nm/mL) treatments, respectively. After incubating at 37 °C for 5 h, graphs were snapped in a 10× field.

### 4.11. Immunohistochemistry (IHC)

After deparaffinization, rehydration, hydrogen peroxide treatment, citrate buffer for antigen retrieval, and goat serum blocking, tumor tissue slides were reserved with specific dilutions of the antibody against Ki67(1:6000, ab15580, Abcam, Cambridge, UK) and CD31(1:3200, #77699, CST, Danvers, Massachusetts, USA) for 8 h at 4 °C. The slides were then incubated with horseradish peroxidase-linked secondary antibody.

### 4.12. Flow Cytometry

The cells were treated with Crizotinib (50 nm/mL) or DMSO for 48 h. Then the Annexin V-FITC Apoptosis Detection Kit (Beyotime, Shanghai, China) was used to detect the level of apoptosis. The rate of apoptosis was tested by a flow cytometer (Beckman Coulter, Brea, CA, USA).

### 4.13. Modeling HCC in Mice Using Subcutaneous Xenografts

Hepa1-6 (2 × 10^6^/150 μL dissolved in 40% Matrigel and PBS per mouse) was injected into the right flank of C57BL/6 mice (5 mice/group). The mice were treated with PBS (100 μL) and Crizotinib (20 mg/kg, 100 μL) daily by gavage. Each subcutaneous tumor was dissected, and its weight was measured every day.

### 4.14. cDNA Synthesis and Quantitative Real-Time PCR (qRT-PCR) Assay

Relative quantitative real-time PCR was used to confirm the expression of 14 ARPGs in HCC cell lines. Trizol reagent (Invitrogen, Waltham, MA, USA) was used to extract total RNA. Total RNA was reverse-transcribed into cDNA according to the instructions for the Hifair^®^ V one-step RT-gDNA digestion SuperMix for qPCR (YEASEN, Shanghai, China). Then, qRT-PCR assays were conducted by the ABI7500 system according to the protocol. The primer sequences for qRT-PCR were listed (Table 1):

### 4.15. Statistical Analysis

The Version 4.1.2 R software was used to process statistical analyses. Unpaired Student’s *t*-test was used to determine statistical significance for regularly distributed variables for comparisons between two groups, and the Mann-Whitney U test was used to analyze non-normally distributed data. Kruskal-Wallis and one-way ANOVA tests for non-parametric and parametric procedures, respectively, were employed for comparisons of more than two groups. Using Spearman and distance correlation analysis, the correlation coefficient was calculated. The log-rank (Mantel-Cox) test was used to identify statistically significant differences, and the Kaplan-Meier method was employed to produce survival curves for the subgroups in each data set. The univariate Cox proportional hazards regression model was used to generate the hazard ratios for single-variate studies. To identify independent prognostic markers, the multivariate Cox regression model was employed. In our work, *p* ≤ 0.05 was deemed to be statistically significant.

## 5. Conclusions

In conclusion, our present study identified genes associated with angiogenesis that were key to HCC prognosis. Additionally, this 14-mRNA predictive signature remained significant even for patients with different subtypes. It appears that this model may contribute to the personalized management of HCC patients.

## Figures and Tables

**Figure 1 ijms-24-03324-f001:**
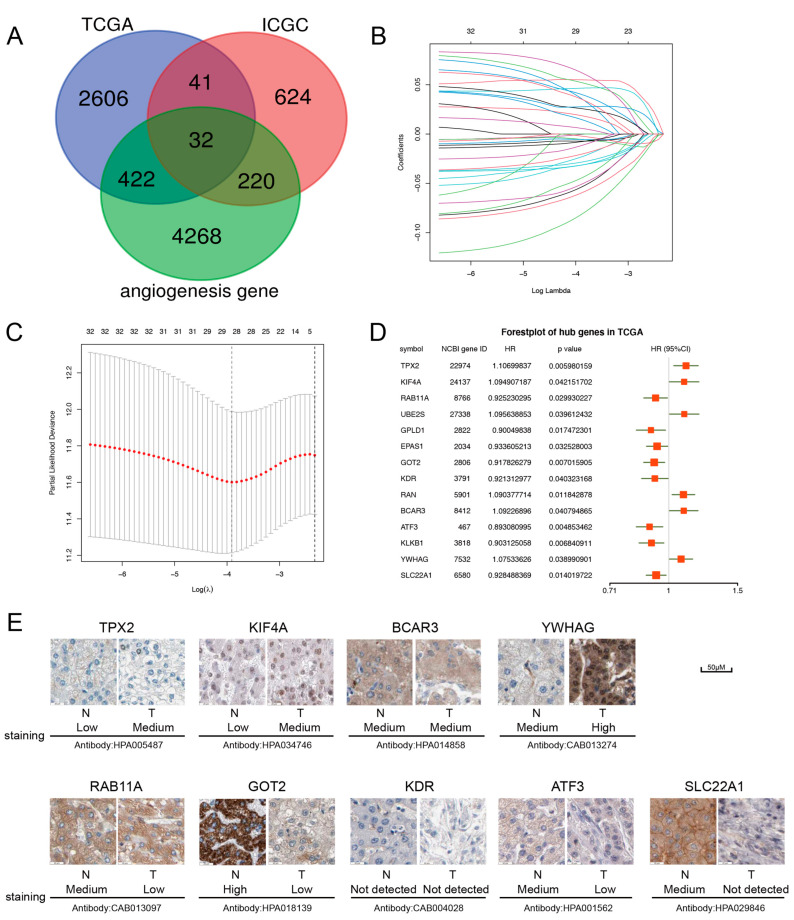
The construction of the prognostic signature. (**A**) Venn diagram to identify angiogenic genes related to survival. (**B**,**C**) The optimal value of λ was selected by LASSO regression. (**D**) The 14 ARPGs were selected by univariate Cox regression analysis. (**E**) Immunohistochemical data of ARPGs in HPA database.

**Figure 2 ijms-24-03324-f002:**
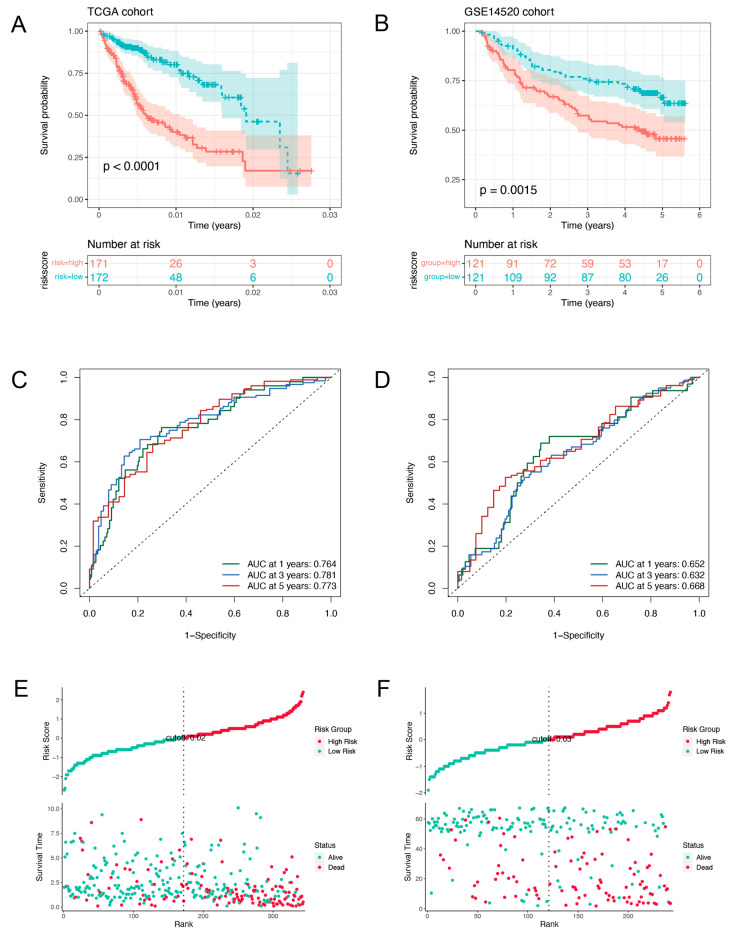
Validation of the risk score model in TCGA (The left column) and GSE14520 (The right column) sets. Kaplan-Meier curves were made based on the ARPG risk model in the training group (**A**) and validation group (**B**). The ROC curves of 1, 3, and 5-year survival for HCC patients in the training group (**C**) and validation group (**D**). HCC patients’ survival status, interval according to their risk score (**E**) and validation group (**F**).

**Figure 3 ijms-24-03324-f003:**
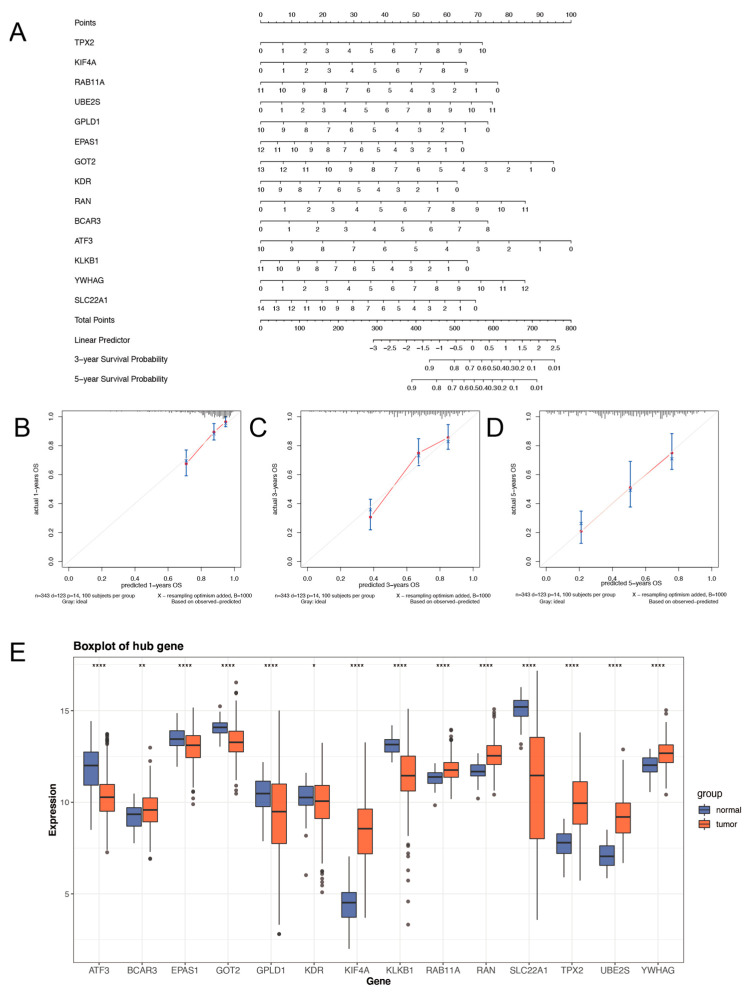
The construction of a nomogram for OS prediction in HCC. (**A**) The nomogram consists of 14-ARPG. (**B**–**D**) Estimating survival rates with the nomogram calibration curves at 1, 3, and 5 years. (**E**) The expression level of the 14-ARPG gene in normal and HCC tissues. * 0.01 < *p* < 0.05; ** 0.001 < *p* < 0.01; **** *p* < 0.0001; ns no significance.

**Figure 4 ijms-24-03324-f004:**
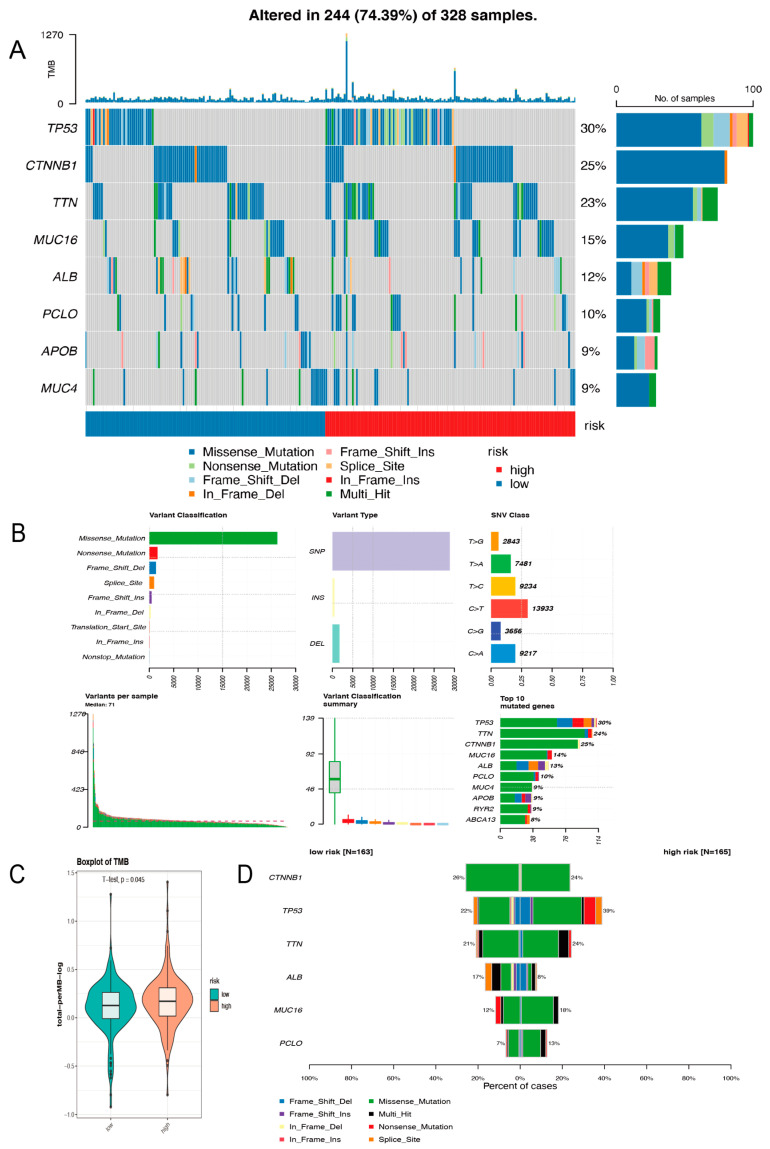
Analyzing somatic variants among different risk groups. (**A**) Top mutated genes in the TCGA cohort. (**B**) A landscape of somatic mutations. (**C**) Differences in tumor mutation burden between two groups. (**D**) A comparison of high-risk and low-risk groups based on the top mutated genes.

**Figure 5 ijms-24-03324-f005:**
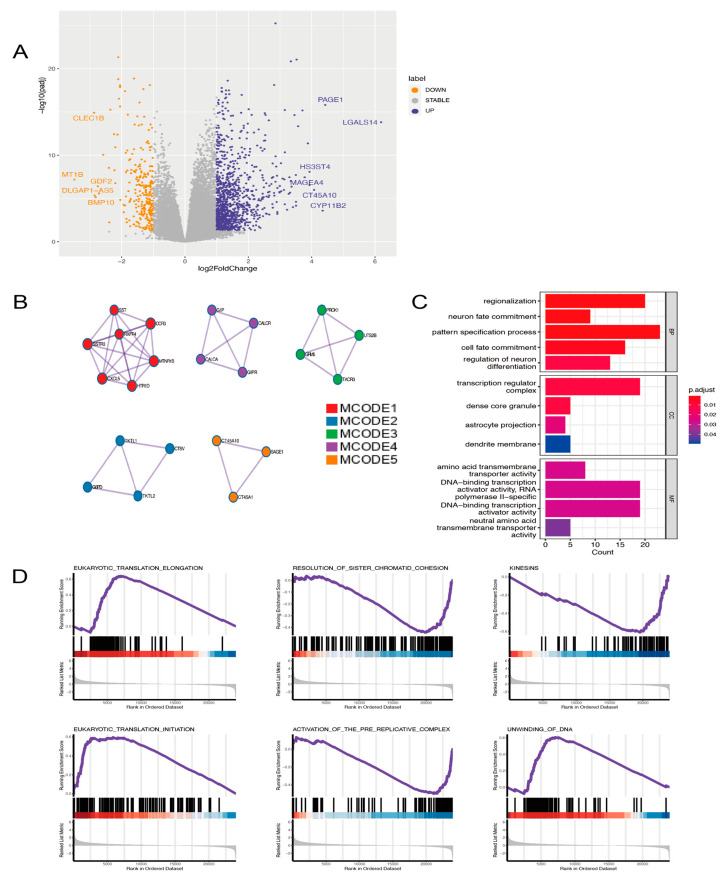
The corresponding regulatory network of ARPGs. (**A**) The volcano plot of DEGs between high- and low groups. (**B**) There is an identification of the protein-protein interaction network and the components of MCODE. (**C**) The first five pathways of GO enrichment in biological processes, cellular components, and molecular functions. (**D**) GSEA analysis results showed that the top six pathways were identified by the Reactome pathway database.

**Figure 6 ijms-24-03324-f006:**
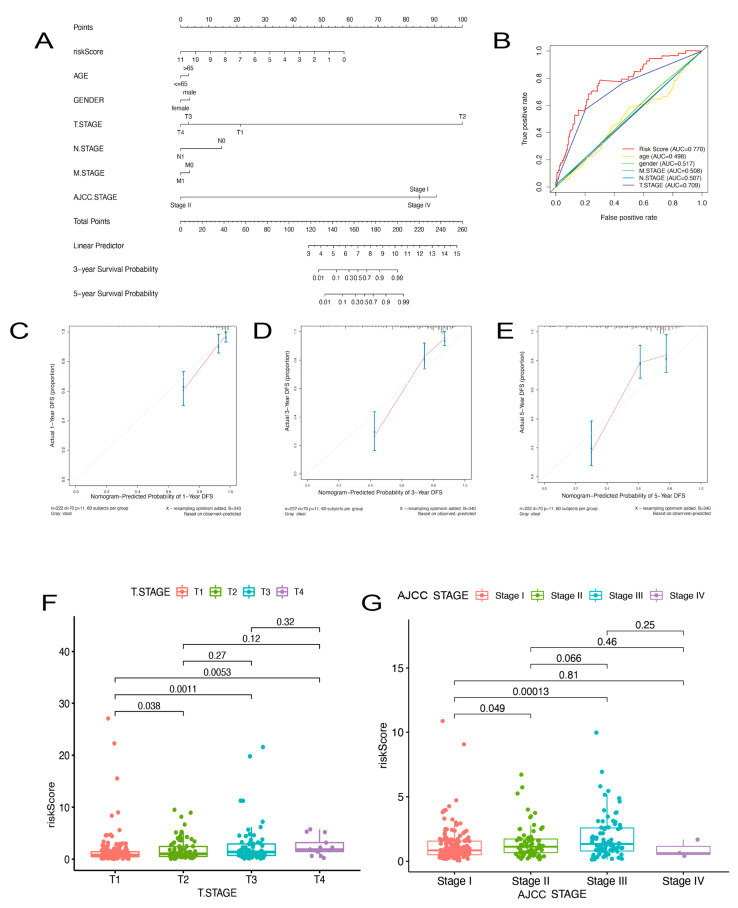
Construction of a nomogram for prediction prognosis. (**A**) A nomogram for predicting OS in HCC patients. (**B**) ROC curve analysis of every variable. (**C**–**E**) Calibration plots of 1, 3 and 5 years for predicting survival for patients. (**F**,**G**) Boxplot of risk score between different HCC stages in the LIHC cohort. The Kruskal-Wallis test was used to verify the signature.

**Figure 7 ijms-24-03324-f007:**
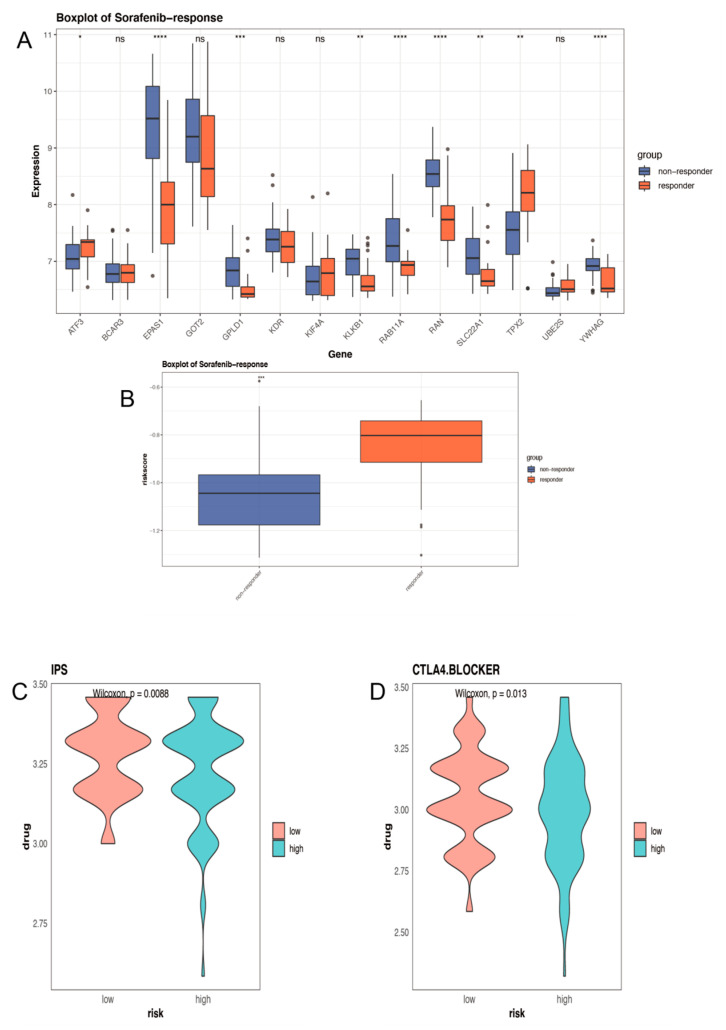
Prediction of treatment response for HCC patients. (**A**) Differences in the expression of ARPGs between responders and non-responders to Sorafenib. (**B**) The difference in risk score between Sorafenib treatment responders and non−responders. (**C**) IPS score distribution plot. (**D**) IPS−CTLA4 blocker score distribution plot. * 0.01 < *p* < 0.05; ** 0.001 < *p* < 0.01; *** 0.0001 < *p* < 0.001; **** *p* < 0.0001; ns no significance.

**Figure 8 ijms-24-03324-f008:**
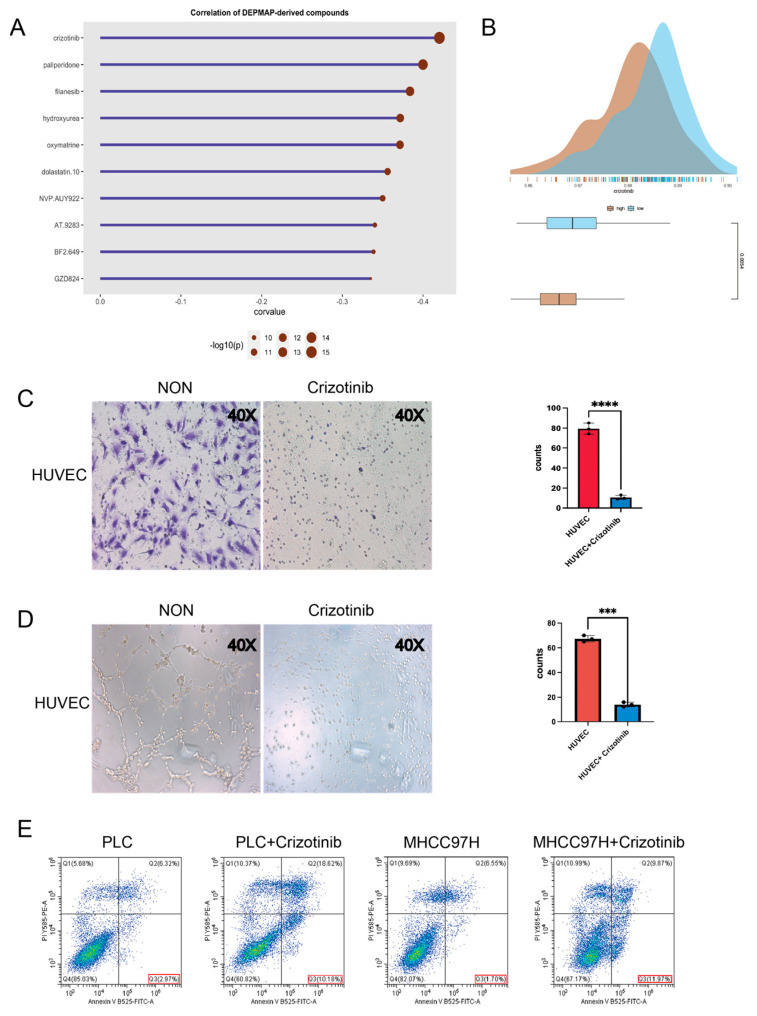
Prediction of potential targets of high risk-group. (**A**) The correlation of DEPMAP−derived compounds was shown in the Cleveland diagram. (**B**) AUC of Crizotinib in different risk groups. (**C**) The transwell assay confirmed the Crizotinib treatment’s ability to migrate HUVECs. (**D**) A Matrigel−based capillary-genesis test was performed on HUVEC tube formation. (**E**) Flow cytometry for testing the apoptosis of HCC cell lines after Crizotinib treatment. *** 0.0001 < *p* < 0.001; **** *p* < 0.0001.

**Figure 9 ijms-24-03324-f009:**
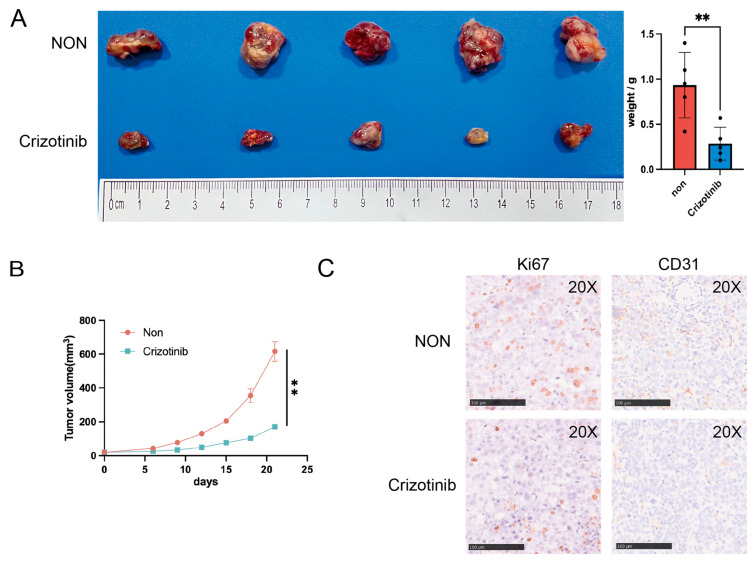
Experiments for Crizotinib treatment of liver cancer in vivo. (**A**) The xenografts tumor from different treatment groups. (**B**) The tumor volumes between the two groups were tracked for 21 days. (**C**) IHC staining for the level of Ki67 and CD31 in tumor between two groups. ** 0.001 < *p* < 0.01.

**Table 1 ijms-24-03324-t001:** Primer sequences for 14 ARPGs.

Symbol	Forward Primer	Reverse Primer
*TPX2*	ATGGAACTGGAGGGCTTTTTC	TGTTGTCAACTGGTTTCAAAGGT
*KIF4A*	TACTGCGGTGGAGCAAGAAG	CATCTGCGCTTGACGGAGAG
*RAB11A*	CAACAAGAAGCATCCAGGTTGA	GCACCTACAGCTCCACGATAAT
*UBE2S*	ACAAGGAGGTGACGACACTGA	CCACGTTCGGGTGGAAGAT
*GPLD1*	ATGTCTGCTTTCAGGTTGTGG	ACGCATCCTGGTGTTCTAGTAA
*EPAS1*	CGGAGGTGTTCTATGAGCTGG	AGCTTGTGTGTTCGCAGGAA
*GOT2*	AAGAGGGACACCAATAGCAAAAA	GCAGAACGTAAGGCTTTCCAT
*KDR*	GGCCCAATAATCAGAGTGGCA	CCAGTGTCATTTCCGATCACTTT
*RAN*	GGTGGTACTGGAAAAACGACC	CCCAAGGTGGCTACATACTTCT
*BCAR3*	CAGAAACATGCCGGTGAATCA	GTGGGGATTTGGAGTGGGG
*ATF3*	CCTCTGCGCTGGAATCAGTC	TTCTTTCTCGTCGCCTCTTTTT
*KLKB1*	TCCTTGTTTGCTACAGTTTCCTG	TCTGGCAGTATTGGGCATTTG
*YWHAG*	AGCCACTGTCGAATGAGGAAC	CTGCTCAATGCTACTGATGACC
*SLC22A1*	ACGGTGGCGATCATGTACC	CCCATTCTTTTGAGCGATGTGG
GAPDH(reference gene)	GGAGCGAGATCCCTCCAAAAT	GGCTGTTGTCATACTTCTCATGG

## Data Availability

The bioinformatics data used in support of the findings of this study are all from public data.

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
