# Peer review of "An Angiogenic Gene Signature for Prediction of the Prognosis and Therapeutic Responses of Hepatocellular Carcinoma"

_ijms, 2023, doi:10.3390/ijms24043324_

Round 1

Reviewer 1 Report

Ci et al. present findings about an angiogenesis-related gene signature predictive of survival in HCC patients. They correlate this score to tumor mutational burden as well as responses to immune checkpoint inhibitors and Sorafenib. Additionally, they examine the effect of an anti-angiogenic drug Crizotinib on tumor growth in murine subcutaneous models.

CRITIQUES

1.       Fig 3E is described as levels of ARPG genes between heathy and cancer patients but the graph is labeled as normal and tumor. Is data comparing tumor vs normal tissue? Or healthy vs cancer patients? Which data was used for this analysis? If this is comparing healthy vs normal, then how was normal liver tissue obtained? Please include detailed protocols and sources.

2.       Fig 4 describes that high-risk group has higher TMB which could predict higher response to ICI. Authors should include data whether any patients in this analysis received immunotherapy, and how does that correlate with risk score and/or survival.

3.       Line 183-185- What surface expression is being referred here? Why is that relevant to that analysis?

4.       What are the expression levels of the ARPG genes in the cell lines/tumors used for the in vivo experiments?

Author Response

We would like to thank you for your kind consideration of our manuscript. We also appreciate the constructive and enthusiastic comments from you. The comments have been most useful for us in the preparation of a more concise manuscript. Accordingly, we have revised the enclosed manuscript based on your suggestions. We have uploaded the response letter in the attachment, which contains all the changes to your comments.

Reviewer 2 Report

The article „An Angiogenic Gene Signature for Prediction of the Prognosis and Therapeutic Responses of Hepatocellular Carcinoma” by Ci et al. reports the identification of a panel of angiogenesis-associated genes signature in HCC which might predict patients’ prognosis and outcome.

Overall, the findings are described very concisely. One criticism is that the figure legends are not precise enough. Font sizes in all figures are too small and illegible, especially when printing this manuscript. Further, some English correction is required.

Comments:

1.     There are many empty spaces in this manuscript including page 2, 4, 7, 9, 11, 13 and 15. Please rearrange the figures to avoid waste of space.

2.     Figure 1D: a) The axis label of the forest plot is illegible. Please choose a taller font size and add the axis description for the vertical line (x = 1?). b) Please add the NCBI gene ID or gene reference sequence per gene. c) Which values are shown? Please add a headline above the values on the left side and add a description of depicted values (red square) and errors (black lines) in the Figure legend 1D.

3.     Figure 1E: a) Please show a higher magnification of the IHC and not the full microarray. It is impossible to see the cellular level of the IHC stainings. b) Please add information on the shown magnification or add a scale bar to the Figure legend 1E.

4.     Page 2, lines 70­­­–71: The TMAs for BCAR3 and GOT2 are not representable since they depict the opposite compared to the results described in the text. Levels of BCAR3 appear to be lower in tumor tissue compared to normal tissue and GOT2 levels higher in tumor tissue than in normal liver. Please doublecheck and replace the IHC images by more representative ones.

5.     Page 4, lines 79–80: Please give a definition for low- and high-risk scores in the materials a methods section.

6.     Figure 2A and B: Please choose a taller font size; axis labels are illegible.

7.     Figure 3A: Please describe in the Figure legend 3A what is shown here.

8.     Figure 3B–D: The text of all graphs is illegible and the lines in the blots are too thin. Please revise all three graphs and make sure the presented data and text are visible.

9.     Page 6, lines 100–101: “Comparison of high- and low-risk groups for somatic variants.” This is no sentence; I assume this is a heading for the next paragraph. Please clarify and move to the correct place in the text.

10.  Page 7, line 107: Please write “showing” instead of “showed”.

11.  Page 7, lines 106–124: What is the rational of mentioning immunotherapy here? What does immunotherapy have to do with the panel of angiogenesis-associated genes discovered in this study? Please clarify and write an explanatory sentence prior to line 106.

12.  Page 7, lines 106–124: Font size is inconsistent. Please correct.

13.  Page 7, lines 113–115: Figure 4 B requires more description. Please add some more text describing the findings.

14.  Page 7, line 124: “Analyzing the model’s functional enrichment”. Again, I assume this is a heading for the next paragraph. Please move this phrase to the correct position.

15.  Figure 5B: Figure 5B is irrelevant due to illegibility. Please remove Figure 5B.

16.  Page 9, lines 133–136: Are the 14 ARPGs included in the panel of 500 genes studied via Metascape? Please clarify in the text.

17.  Figure 5E–J: Revise these graphs. Nothing is legible here.

18.  Figure 6A: Please describe in the Figure legend 6A what is shown here.

19.  Figure 6F and G: Which statistical test was applied here. Please mention the test in the Figure legend 6F and G.

20.  Page 12, line 180: Write “differently expressed” instead of “expressing differently”.

21.  Page 15, line 206: Please explain in the text why HUVEC cells were used for migration and tube formation assays.

22.  Figure 8E: What does 97h mean? In the methods section, the authors claim that cells were treated for 48 h with crizotinib. Please clarify.

23.  Figure 9C: Please choose higher magnification images of KI67 and CD31 IHC. Add a scale bar or magnification.

24.  Page 19, line 311: The authors grouped the HCC patients into low- and high-risk subgroups. Please add the definition of low- and high-risk, respectively.

25.  Page 20, line 349: Why did the authors use premature 3 to 4-weeks old animals? Usually, for xenograft experiments, adult animals of earliest age of 6-7 weeks are used. Please explain.

26.  Page 20, line 353: Please add the protocol number and date of animal approval.

27.  Page 20, lines 368–369: Please add the antibodies used for IHC including order number and dilution.

28.  Page 20, line 376: Please write “injected” instead of “shot”.

29.  Page 20, line 380–382: Please provide more information on the statistical analyses that were applied.  

Author Response

(The authors gave the same response as above.)

Round 2

Reviewer 2 Report

Dear authors,

Please find attached reviewer 2's responses. The reviewer's comments have been addressed to nearly full satisfaction. Please pay regard to point 26 and 27 and implement these information in the final revised version of the manuscript. 

Kind regards

Author Response

We appreciate your efficient work efficiency in reviewing our manuscript. Thank you for your recognition of our work and your valuable comments. We have added the information to our newest manuscript. The added content and more details are submitted by us in the attachment.

Sincerely. 

Response to Reviewer 2 Comments

We would like to thank you for your kind consideration of our manuscript entitled “An Angiogenic Gene Signature for Prediction of the Prognosis and Therapeutic Responses of Hepatocellular Carcinoma” (ijms-2160053). We appreciate your efficient work efficiency in reviewing our manuscript. Thank you for your recognition of our work and your valuable comments.

We have added the information to our newest manuscript:

Response:

Line 377-380: After deparaffinization, rehydration, hydrogen peroxide treatment, citrate buffer for antigen retrieval, and goat serum blocking, tumor tissue slides were reserved with specific dilutions of the antibody against Ki67(1:6000, ab15580, Abcam) and CD31(1:3200, #77699, CST) for 8h at 4 °C.

Line 427-428: Institutional Review Board Statement: The animal study was approved by the Department of Laboratory Animals Science, Fudan University (2018华山医院JS-009) on 2018.02.27.